# Human Infections by *Wohlfahrtiimonas chitiniclastica*: A Mini-Review and the First Report of a Burn Wound Infection after Accidental Myiasis in Central Europe

**DOI:** 10.3390/microorganisms9091934

**Published:** 2021-09-11

**Authors:** Martin Hladík, Bretislav Lipovy, Yvona Kaloudova, Marketa Hanslianova, Ivana Vitkova, Tereza Deissova, Tomas Kempny, Martin Svoboda, Zdenek Kala, Pavel Brychta, Petra Borilova Linhartova

**Affiliations:** 1Department of Burns and Plastic Surgery, Faculty of Medicine, Institution Shared with University Hospital Brno, Masaryk University, Jihlavská 20, 625 00 Brno, Czech Republic; hladikm@gmail.com (M.H.); bretalipovy@gmail.com (B.L.); kaloudova.yvona@fnbrno.cz (Y.K.); tomas.kempny@gmail.com (T.K.); dr.brychta@seznam.cz (P.B.); 2CEITEC–Central European Institute of Technology, Brno University of Technology, Purkyňova 656/123, 612 00 Brno, Czech Republic; 3Department of Clinical Microbiology, Vyškov Hospital, Purkyňova 235/36, 628 01 Vyškov, Czech Republic; hanslianova.marketa@nemvy.cz; 4Department of Clinical Microbiology and Immunology, University Hospital Brno, Jihlavská 20, 625 00 Brno, Czech Republic; vitkova.ivana@fnbrno.cz; 5Environmental Genomics Research Group, RECETOX, Faculty of Science, Masaryk University, Kamenice 5, 625 00 Brno, Czech Republic; tdeiss@mail.muni.cz; 6Department of Surgery, Faculty of Medicine, Institution Shared with University Hospital Brno, Masaryk University, Jihlavská 20, 625 00 Brno, Czech Republic; svoboda.martin2@fnbrno.cz (M.S.); kala.zdenek@fnbrno.cz (Z.K.); 7Clinic of Maxillofacial Surgery, Faculty of Medicine, Institution Shared with the University Hospital Brno, Masaryk University, Jihlavská 20, 625 00 Brno, Czech Republic

**Keywords:** *Wohlfahrtiimonas chitiniclastica*, burn wound infection, myiasis

## Abstract

*Wohlfahrtiimonas chitiniclastica* are bacteria that cause rare infections, typically associated with the infestation of an open wound with fly larvae. Here, we present a unique case report of the first *W. chitiniclastica* isolation from a burn wound with accidental myiasis in a 63-year-old homeless man and a literature review focused on human infections caused by these bacteria. So far, 23 cases of infection with *W. chitiniclastica* have been reported; in 52% of these, larvae were found in the wound area. Most of these cases suffered from chronic non-healing wound infections but none of these were burn injuries. The overall fatality rate associated directly with *W. chitiniclastica* in these cases was 17%. Infections with parasitic larvae occur in moderate climates (especially in people living in poor conditions); therefore, an infection with rare bacteria associated with accidental myiasis, such as *W. chitiniclastica*, can be expected to become more common there. Thus, in view of the absence of recommendations regarding the treatment of patients with accidental myiasis and, therefore, the risk of infection with *W. chitiniclastica* or other rare pathogens, we provide a list of recommendations for the treatment of such patients. The importance of meticulous microbial surveillance using molecular biological methods to facilitate the detection of rare pathogens is emphasized.

## 1. Introduction

*W**ohlfahrtiimonas chitiniclastica* was first described in 2008 by Tóth et al. [1]. These bacteria are strictly aerobic gram-negative, straight short rods (1.5–2.0 × 0.5–1.0 μm) that are non-motile, non-spore-forming, which grow at pH 5.0–10.5 and temperatures of 28–37 °C, and are capable of causing both local skin/soft tissues infections (SSTIs) and sepsis. *W. chitiniclastica* is catalase and oxidase-positive, and indole, urease and H_2_S negative. It has a strong chitinase activity, which leads to discussions about whether or not its presence can be symbiotic and play a role in the larval metamorphosis [1,2].

The presence of *W. chitiniclastica* was associated with fly larvae infestation in open wounds; nevertheless, an infection with this bacterium is very rare in humans [2]. In the last decade, rare pathogens have begun to emerge as causes of the development of infectious complications. To date, there is no report about a *W. chitiniclastica* infection in a burn wound [3].

This paper describes the first isolation worldwide of *W. chitiniclastica* from a burn wound with accidental myiasis (interestingly in a patient from Central Europe) and reviews available information on *W. chitiniclastica*-caused infections in humans. In addition, a set of recommendations for care of a patient with accidental myiasis and, therefore, a possible infection with *W. chitiniclastica* and other rare pathogens, is proposed.

## 2. Case Report

We describe the case of a 63-years-old man who was transferred to the Department of Burns and Plastic Surgery of the University Hospital Brno, Czech Republic, with deep burns in the region of thorax and abdomen, covering 5% of the total body surface area (TBSA). His vitals on admission showed a body temperature of 37.2 °C, blood pressure of 163/87 mmHg, and a heart rate of 98/min. On admission, the patient suffered from massive cutaneous myiasis, pediculosis of the head, and phlegmon of the thorax. The patient lived in poor and unhygienic conditions (homeless/staying in shelters for homeless people overnight). He admitted to tobacco and alcohol abuse; he also had hepatitis A in the past. He reported that his burns were the result of an injury that happened two weeks prior, when his shirt caught fire while smoking a cigarette. Right after the injury, first aid was provided by the shelter employee. For the next two weeks, however, the wound was left unattended. The wound showed signs of accidental myiasis, redness, and edema (Figure 1).

The maggot infestation was immediately mechanically removed, and swabs were taken for microbiological analysis. Local therapy included Octenisept^®^ (Schülke & Mayr GmbH, Norderstedt, Germany) for antisepsis and Flamazine^®^ cream 1% *w*/*w* (Smith & Nephew Pharmaceuticals Ltd., Hessle Road, Hull, UK) for wound-bed preparation; at the same time, systemic oral therapy with amoxicillin, clavulanic acid (1 g every 8 h) and metronidazole (500 mg every 8 h) was initiated. Dressings were changed every 2 days; no maggots were observed after the first infestation removal and the wound bed was prepared for closure. *Staphylococcus aureus*, *Streptococcus pyogenes*, and *W. chitiniclastica* were isolated from the initial swab, all susceptible to amoxicillin and clavulanic acid. On day 7, debridement under general anesthesia was performed and defects were finally covered with split-thickness skin grafts on 3% TBSA (Figure 2).

The next series of swabs revealed only coagulase-negative *Staphylococcus* in a single spot; hence, the systemic therapy with amoxicillin and clavulanic acid continued for 8 days. All skin defects were completely healed, and the patient was discharged to our outpatient care and long-term care ward for intensive rehabilitation (Figure 3).

### 2.1. Pathogen Identification

As a part of the microbiological examinations, swabs from the patient’s burn areas were transferred to culture media suitable for both aerobic and anaerobic growth; blood agar (Merck KGaA, Darmstadt, Germany), McConkey agar (selective medium for growth of gram-negative bacteria; Pronadisa, Conda Laboratories, Madrid, Spain) and blood agar with NaCl addition (selective agar for staphylococci; Merck KGaA, Germany) were used for aerobic culture and VL agar (Bio-Rad Laboratories Chemical Division, Richmond, CA, USA) was used for anaerobic culture. The inoculated culture media were then incubated for 18–24 h at 35–37 °C; blood agar and chocolate agar (OXOID CZ, Brno, Czech Republic) in the atmosphere with elevated CO_2_ concentration, VL agar in the anaerobic atmosphere, and the remaining media in a normal atmosphere. The growth of the *W. chitiniclastica* strain on the culture media and the morphology of the individual colonies are shown in Figure 4.

The colonies were then identified using the MALDI-TOF Biotyper (Bruker Daltonics, GmbH, Leipzig, Germany) and software MALDI TOF MS, version 3.1.

In addition, bacterial DNA obtained from the culture of *W. chitiniclastica* was isolated by the QIAamp BiOstic Bacteremia DNA Kit (QIAGEN Redwood City, CA, USA). DNA was analyzed by the 16S rRNA Sanger sequencing method (SEQme s.r.o., Dlouha 176 26,301 Dobris, Czech Republic), forward primer 5′-AYTGGGYDTAAAGNG-3′ and reverse primer 500B4-CCGTCAATTYYTTTRAGTTT-3′ were used. A 98% match of the specified sequence with the sequence of *W. chitiniclastica* DSM 18,708 strain S5 16S rRNA was found using the basic local alignment search tool (BLAST) provided by the National Center for Biotechnology Information (NCBI) [4] (Figure 5).

### 2.2. Antimicrobial Susceptibility Profile of W. chitiniclastica

The susceptibility of *W. chitiniclastica* to antibiotics was established by determining the minimum inhibitory concentration using the MicroScan WalkAway MIC instrument (Beckman Coulter, Brea, CA, USA). Susceptibilities were evaluated using a disc diffusion method according to the EUCAST standard–PK/PD (non-species related) breakpoints (EUCAST Clinical Breakpoints Tables, v.10.0, January 2020). The *W. chitiniclastica* strain was sensitive to the tested antibiotics, including the amoxicillin/clavulanate used in the therapy (Table 1).

## 3. Review and Discussion

Myiasis (from the Greek myia, which means fly, and iasis, disease) is defined as an infestation of humans or other vertebrates with a larval stage of an insect. The term was first used by F. W. Hope in 1840, who used it to distinguish diseases caused by the larvae of dicotyledons from other parasitic insects [5]. Such larvae are usually detected in necrotic tissue (benign myiasis), but can also affect fully viable tissue (malignant myiasis). In the past, myiasis was almost exclusively described in tropical or subtropical climates [6]. In recent years, it has also been found in temperate climates [6].

Two principal types of myiasis can be distinguished, controlled and accidental. Controlled myiasis is a therapeutic procedure using sterile larvae for selective debridement of necrotic tissue and facilitation of the wound-healing phase. This methodology is mainly used in the treatment of non-healing wounds. Larvae can also be used for reducing the bioburden of gram-positive strains of bacteria. In modern times, this method was first approved for use in 2004 by the Food and Drug Administration [7].

Accidental myiasis usually arises as a manifestation of parasitic larvae in non-healing wounds in high-risk groups of patients (alcohol addiction, homelessness, poor hygiene, tobacco use, chronic vascular diseases, immunocompromisation, non-compliant patients, low socioeconomic status, etc.). Our patient had almost all of these risk factors (he was a homeless, alcohol and tobacco abuser with poor living conditions). His burns were not treated and covered with any dressings for 2 weeks prior to hospitalization (hence the notion of non-compliancy). In accidental myiasis, the larvae are not sterile and, therefore, can act as a vector for a variety of potentially pathogenic microorganisms, such as *W. chitiniclastica* or *Ignatzschineria indica*.

Accidental myiasis causing bacteremia or sepsis can be potentially lethal. So far, there is no uniform therapeutic concept or methodology for sanitizing the primary larval burden. In the case of cavital myiasis, surgical intervention involving incision and drainage, or the excision of infected tissue may be necessary. In the case of cutaneous myiasis, manual debridement with removal of the larvae is needed; in the case of deeper penetration with larvae, their primary asphyxiation with petroleum jelly or similar material is a suitable method for their eradication.

Burn patients are highly susceptible to infections in the area of the skin defect; infectious complications account at present for the majority of mortality and morbidity in these patients. Immediately after thermal trauma, any burn area is sterile as a result of the heat on the superficial microorganisms. However, the necrotic tissue in the wound bed represents an excellent growth medium for microorganisms, especially for bacteria with a short generation time. Burn wound infections are relatively diverse, ranging from cellulitis and impetigo to invasive wound infections requiring prompt antimicrobial treatment with early debridement [8]. In our patient, we used manual debridement with the removal of larvae and antiseptic treatment.

In the course of thermal trauma therapy, both quantitative and qualitative changes in microbial colonization occur in the burn area (as well as in the other parts of the body). While etiologically, gram-positive cocci (*S. aureus*, coagulase-negative *Staphylococci*, *Streptococcus* sp., etc.) predominated the burn wound in the first week of hospitalization, the predominance of gram-negative rods (mainly representatives of *Enterobacteriacae* sp., *Pseudomonas aeruginosa*, *Acinetobacter baumannii*, etc.) begins to manifest from the second week of hospitalization onwards [9]. Rare viral, bacterial, and yeast infections in burn patients occur mostly as a result of the encounter with microorganisms of exogenous origin [10]. The burn wound infection in our patient with minor burns was of polymicrobial etiology; interestingly, *W. chitiniclastica* was cultured from the burn wound, which was also confirmed by the sequencing analysis. It should be mentioned that *W. chitiniclastica* was cultured on blood agar in an elevated CO_2_ concentration, which falls within a standard set of culture conditions; therefore, no special culture is necessary.

*W.**chitiniclastica* can be found in *Wohlfahrthia magnifica* (Diptera, Sarcophagidae) a parasitic flesh fly native to continental Europe, Asia and the Mediterranean, *Lucilia sericata* (green bottle fly) in the Americas, *Chrysomya megacephala* (oriental latrine fly), *Hermetia illucens* (black soldier fly), and *Musca domestica* L. (housefly) [7,11,12]. Although most reports of *W. chitiniclastica* infections have been associated with the larvae of parasitic flies, this bacterium has also been isolated in retail frozen chicken meat, arsenic-affected soils in Bangladesh, and in cow [13,14,15].

Despite the fact that *W. magnifica* is the most likely vector of this myiasis in the Czech Republic, another possible vector, a new invasive representative of the order Diptera, *Clogmia albipunctata*, has been also reported in the geographical area where our patient resided [16]. The identification of the vector was not performed in our case. Although it is not important for clinical practice, it may be perceived as a limitation of this paper from the epidemiological perspective. If a particular vector overpopulation occurs in a certain area, we can potentially expect an increased risk of rare pathogen infections, including *W. chitiniclastica*. For this reason, we would also like to recommend the determination of the vector, where possible, in any future cases of myiasis, especially if a rare pathogen is identified. This can be performed, for example, by inserting the larvae into a fixation solution and if a rare pathogen is determined, cooperation with entomologists can be initiated.

Most reports of *W. chitiniclastica* infections originate from subtropical or tropical regions [2,11,17,18,19,20,21,22,23,24,25,26,27,28,29,30,31,32] (see the literature review in Table 2). Nevertheless, an infection with bacterium was also reported from northern Europe [18]. Similarly, our case did not occur in subtropical or tropical conditions either, but in a moderate climate in the autumn with relatively low day temperatures (the mean temperature at our patient’s place of residence between the accident and hospitalization was 14.13 ± 3.75 °C). This supports the notion that living conditions are extremely important for the development of *W. chitiniclastica.*

The spectrum of infections in the total of the 23 reported patients ranges from wound infection (43% of all reported patients) to bacteremia or bloodstream infection (61% of all reported patients). In some cases, the infection might have resulted in septic shock; however, we are not aware of any reported case of burn wound or other acute wound infection. Most reported cases described secondary infections in patients with non-healing wounds (34% of patients reported). Although infection with *W. chitiniclastica* is generally considered to be associated with maggot infestation, no larvae were identified in the wound area in 48% of the described cases.

The mortality associated with this type of infectious complication depends on the host condition and immunological status, but also on the level of invasiveness/severity of the infection itself and, of course, the quality of care. The overall fatality rate reported in previous studies was 26%; however, death clearly associated with *W. chitiniclastica* infection was reported only in 17% of patients (all of whom had an invasive infection). Only six patients (26%) described in the literature suffered from a monobacterial *W. chitiniclastica* infection; the remaining 17 (74%) were infected with multiple pathogens. This was also the case of our patient who presented with multiple pathogens (besides *W. chitiniclastica*, *S. aureus*, and *S. pyogenes* were also detected in the burn wound).

Since infection with *W. chitiniclastica* is very rare, no standardized antimicrobial protocol for its treatment has been established so far. The most commonly used antibiotics in the treatment of such infections include beta-lactams (mainly amoxicillin-clavulonate, cefuroxime, ceftriaxone, or cefepime) or fluoroquinolones (ciprofloxacin, levofloxacin).

The situation is further complicated by the fact that, as mentioned above, many of the cases described in the literature are not monopathogenic infections; therefore, all pathogens present in the wound must be taken into consideration. Therefore, we recommend the following when encountering a patient with accidental myiasis (and with an increased possibility of infection with *W. chitiniclastica* or other rare pathogens):Larvae should be removed from the wound (cutaneous myiasis); non-invasive removal of larvae is possible by occlusion or suffocation using petroleum jelly, beeswax, or liquid paraffin. Over the course of several hours, the larvae either leave the wound or die and can be removed more easily from the wound [33]. Surgical removal of larvae requires repeated debridement with an application of antiseptic agents. In the case of cavitary myiasis, surgical debridement, incision or even excision of the infected tissue may be necessary [6]. The larvae should be also preserved for potential future identification;Microbiological analysis focusing on the bacteria typically associated with accidental myiasis should be performed (e.g., *Proteus mirabilis*, *Ignatzschineria indica*, *Providencia rettgeri*, *Morganella morganii*, and *Staphylococcus aureus*). For the identification of rare pathogens, especially *W. chitiniclastica*, MALDI-TOF MS or 16S rRNA sequencing are recommended [17,22,24] because the use of the VITEK 2 system often leads to misclassification: for example, *W. chitiniclastica* was incorrectly classified as *Acinetobacter lwoffii*, *Comamonas testosteroni* or *Rhizobium radiobacter* [18,20,25]. As no data are available on the performance of other systems, a study evaluating the performance of other methods with respect to the identification of *W. chitiniclastica* can be valuable for clinical practice;The choice of antimicrobial therapy should be tailored to the overall microbiological findings; management should also be based on the severity of the infectious complication.

## 4. Conclusions

Accidental myiasis is no longer the domain of only subtropical or tropical regions; with climate change, the likelihood of its occurrence, especially in people with low socioeconomic status, increases in temperate climates as well. In individuals with unsterile parasitic larvae infestation, infection by rare pathogens, such as *W. chitiniclastica*, can be suspected. To minimize the impact of potential septic complications on the patient, the rapid and accurate identification of microorganisms using molecular-biological methods, followed by effective antimicrobial therapy, is necessary.

## Figures and Tables

**Figure 1 microorganisms-09-01934-f001:**
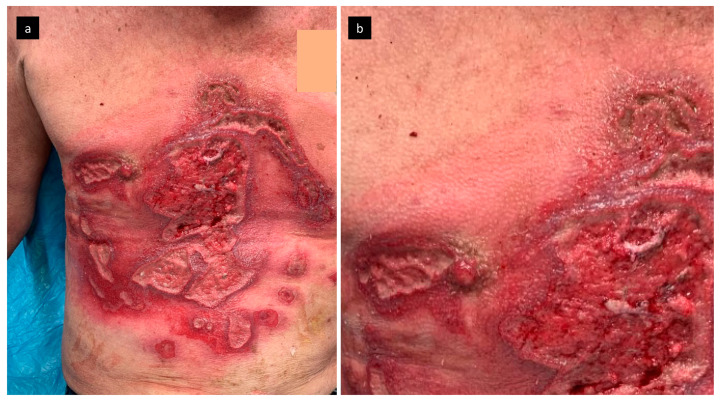
The wound on admission after removal of maggots with signs of redness and edema (**a**), in detail (**b**).

**Figure 2 microorganisms-09-01934-f002:**
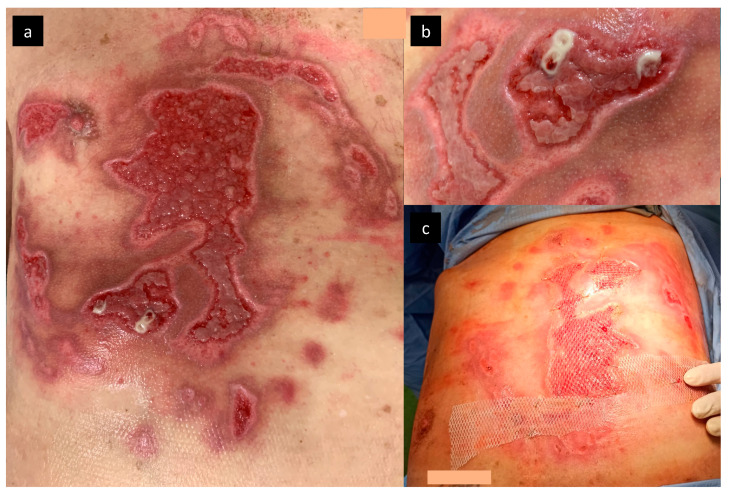
Wound bed preparation with granulation tissue (**a**,**b**), the process of split-thickness skin graft transplantation (**c**).

**Figure 3 microorganisms-09-01934-f003:**
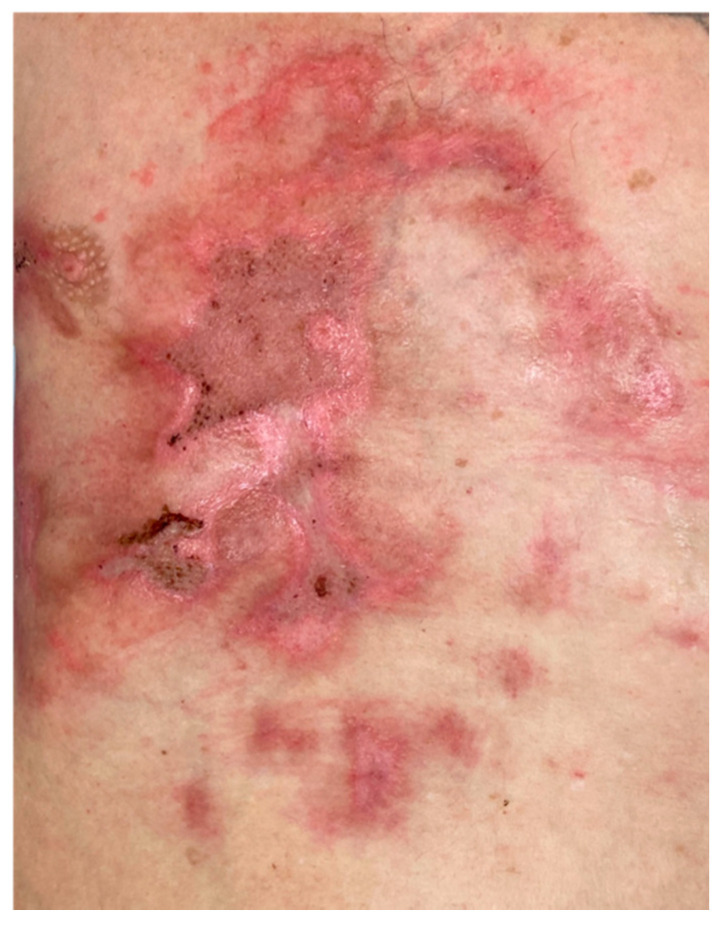
Complete burn wound closure on the discharge of the patient.

**Figure 4 microorganisms-09-01934-f004:**
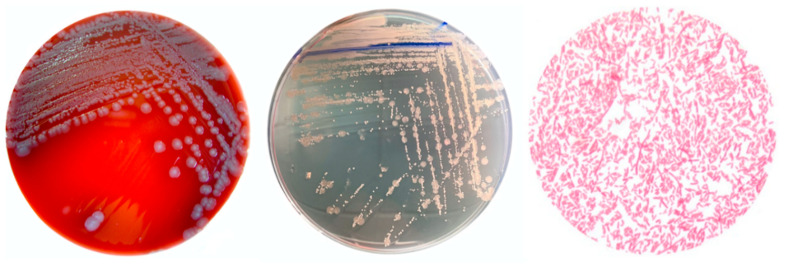
*Wohlfahrtiimonas chitiniclastica* colonies on blood agar (Merck KGaA, Germany) and non-selective chromogenic agar (UriSelect, Bio-Rad Laboratories, Hercules, CA, USA) (**left**, **center**). Microscopic view of the morphology of *W. chitiniclastica* (**right**). Olympus BX40 microscope, Olympus Czech group, s.r.o., magnification 100×. The specimen is stained according to Gram.

**Figure 5 microorganisms-09-01934-f005:**
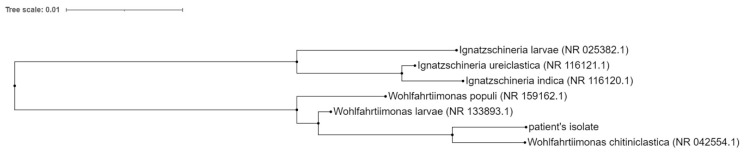
Neighbor-joining phylogenetic tree of partial 16S rRNA gene sequences. Sequences from GenBank are shown with taxonomic names (sequence ID). Query_57927 is the sequence of the patient’s isolate.

**Table 1 microorganisms-09-01934-t001:** Susceptibility of the isolated *Wohlfahrtiimonas chitiniclastica* according to the EUCAST Clinical Breakpoints.

Antibiotic	MIC Value (mg/L)	Interpretation
Amoxicillin/clavulanic acid	<2.0	S
Piperacillin/tazobactam	<4.0	S
Cefotaxime	1.0	S
Ceftazidime	<0.5	S
Cefepime	<0.5	S
Ciprofloxacin	0.25	S
Levofloxacin	0.5	S
Ofloxacin	0.25	S
Imipenem	<1.0	S
Meropenem	<0.5	S

S: sensitive; MIC: minimum inhibition concentration.

**Table 2 microorganisms-09-01934-t002:** Review of the literature of case reports; human infections caused by *Wohlfahrtiimonas chitiniclastica*.

Year [Ref.]	Country	Sex/age Years	Culture Site	Anatomic Localization	Associated Myiasis	Antibiotics	Outcome	Associated Bacteria
2009 [2]	France	F/60	Blood culture	Scalp(head)	Yes	ceftriaxon	Survived	-
2011 [17]	Argentina	M/70	Blood culture	Inguinal regions	No	ciprofloxacin + ampicillin-sulbactam and later ceftazidime + amikacin	Died	
2014 [18]	Estonia	M/64	Resected bone	Right foot	No	amoxicillin-clavulonate	Survived	MYOD
2015 [19]	India	M/43	Ulcer swab	Right lower limb ulcer	No	cefoperazone-sulbactam followed by cefpodoxime (outpatient care)	Survived	
2015 [20]	USA	M/26	Ulcer swab	Right leg	No	cefpodoxime	Survived	PRVU, KLPN, MSSA
2015 [21]	UK	F/82	Blood culture	Excoriations of head, face, and neck	Yes	cefuroxime, metronidazole, clarithromycin, topical chloramphenicol, fucidic acid followed by flucloxacillin (outpatient care)	Survived	PRMI, PRRE, MSSA
2016 [22]	USA	F/69 M/72	F/Ulcer swab M/blood culture	F/Sacral decubitus M/Right foot and umbilical wound	F/No M/Yes	F/ceftaroline fosamil, meropenem, amoxicillin-clavulonate (outpatient care) M/piperacilline-tazobactam, vancomycin, clindamycin	F/Survived M/Died	F/Blood culture: ANSU Urine: PRMI Decubitus: MSSA, AEspp., STspp., BAFR M/ESCO
2016 [23]	South Africa	M/17	Blood culture	Right shoulder wound	No	ceftriaxone	Survived	
2017 [24]	Germany	F/78 M/43 M/71 M/79	F/78 Swabs from ulcers M/43 Swabs from ulcers M/71 Swabs from ulcers M/79 Swab from ulcer	F/78 Leg ulcer M/43 Leg ulcer M/71 Leg ulcer M/79 Leg ulcer	No No No No	F/78 no antibiotic treatment M/43 no antibiotic treatment M/71 no antibiotic treatment M/79 cefuroxime followed by levofloxacin and clindamycin	F/78 Survived M/43 Survived M/71 Survived M/79 Survived	F/78 PRMI, MSSA, SEMA, MOMO M/43 PRMI M/71 PRMI, PRST, PSAE M/79: ESCO
2017 [25]	USA	F/41	Blood culture and decubitus swab	Ischial decubitus or lower extremity bilateral excoriations	No	vancomycin, cefepim, metronidazole	Died due to other disease	Blood culture: PRMI Swab: MYIN, ENFA ENFA, BAspp.
2017 [11]	Malaysia	F/47	Blood culture	-	No	cefoperazone	Died due to other disease	
2018 [26]	USA	M/37	Blood culture	Left lower extremity ulcer	Yes	piperacilline-tazobactam, vancomycin, clindamycin later cefepime	Survived	PRST, IGIN
2018 [27]	Japan	M/75	Blood culture	Left shoulder lesion	Yes	vancomycin, cefepime, metronidazole	Survived	MOMO, STspp., BAspp., PRMI
2018 [28]	USA	M/57	Blood culture	Right ankle gangrene	Yes	Not stated	Died	PRAC, CNS
2019 [29]	Australia	M/54	Blood culture	Right foot wound	Yes	piperacilline-tazobactam + meropenem, ciprofloxacin (outpatient care)	Survived	MOMO
2019 [30]	USA	M/63 F/87	M/blood culture F/swab	M/Right foot ulcer F/Left lower extremity wound	Yes Yes	M/vancomycin+ piperacilin/tazobactam F/unknown	M/Died F/unknown	M/PRRE F/-
2020 [31]	USA	M/82	Blood culture	Right lower extremity	Yes	vancomycin and cefepim, later daptomycin and ceftriaxon	Survived	IGIN, MRSA
2021 [32]	USA	M/70	Blood culture	Ulcer of left temporal region	Yes	levofloxacin	Survived	MSSA, PRMI

MYOD: *Myroides odoratimimus*, PRVU: *Proteus vulgaris*, KLPN: *Klebsiella pneumoniae*, MSSA: *Staphylococcus aureus*, PRMI: *Proteus mirabilis*, PRRE: *Providencia rettgeri*, ANSU: *Anaerobiospirilum succinicproducens*, AEspp.: *Aeromonas* spp., BAFR: *Bacteroides fragilis*, ESCO: *Escherichia coli*, SEMA: *Serratia marsescens*, MOMO: *Morganella morganii*, PSAE: *Pseudomonas aueruginosa*, MYIN: *Myroides injenensis*. ENFA: *Enterococcus faecalis*, BAspp.: *Bacteroides* spp., PRAC: *Propionibacterium acnes*, CNS: *coagulase negative staphylococci*, PRST: Providencia stuartii, IGIN: *Ignatzschineria indica*, MRSA: methicillin-resistant *Staphylococcus aureus*, STspp.: *Streptococcus* spp.

## Data Availability

Data supporting presented results are available upon request.

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
