# Peer review of "Human Infections by Wohlfahrtiimonas chitiniclastica: A Mini-Review and the First Report of a Burn Wound Infection after Accidental Myiasis in Central Europe"

_microorganisms, 2021, doi:10.3390/microorganisms9091934_

Round 1

Reviewer 1 Report

Review on: „Human infections by Wohlfathiimonas chitiniclastica: A mini-review and the first report of a burn-wound infection after accidental myiasis“ by Hladik et al.

General Comments:

The manuscript is well structured and describes for the first time an infection of a burn patient caused by Wohlfathiimonas chitiniclastica, a rare human pathogen. The manuscript presented here is valuable, as case reports make a crucial contribution to the understanding of pathogenicity of rare human pathogenic bacteria. I support the publication, but a few minor improvements need to be made beforehand.

Minor points:

  • Abstract: I would be very careful about relegating the origin of Wohlfathiimonas chitiniclastica infections to the tropics. The first reports derive from more temperate zones and it is unclear whether the bacterium is ubiquitous. The first description was made just over ten years ago ….
  • Introduction: „In the last decade …“ please provide a reference here.
  • Is there evidence to which genus did the maggots belong? Was this examined? Please comment on this and evaluate why this may be an important information in myiasis and Wohlfathiimonas chitiniclastica related infections.
  • Pathogen identification: Please mention the manufacturers of the culture media used here.
  • Pathogen identification: Why were the strains incubated at an elevated CO2 concentration?
  • Pathogen identification: „ …. were then identified using the MALDI TOF ….“ It should be „MALDI biotyper“ or more precise MALDI TOF MS. Please mention the exact unit designation and the versions of software and databases.
  • Pathogen identification: „ … disc diffusion method and minimum inhibition concentrations (MIC)…“ Please mention in this (material and methods) section the manufactures and tests used.
  • Pathogen identification: Please indicate the sequence of the primers used.
  • Review and discussion: „In the past myiasis ….“ Please provide a reference here.
  • Review and discussion: Please speculate whether biochemical methods for identification of Wohlfathiimonas chitiniclastica are useful or not. There is data on VITEK 2 from biomerieux but hardly any data on other systems. Would a follow-up study investigatig other instruments be useful?

Author Response

Specific changes made in manuscript ID: microorganisms-1373594 - revision

To: Reviewer 1

Comment 1: The manuscript is well structured and describes for the first time an infection of a burn patient caused by Wohlfathiimonas chitiniclastica, a rare human pathogen. The manuscript presented here is valuable, as case reports make a crucial contribution to the understanding of pathogenicity of rare human pathogenic bacteria. I support the publication, but a few minor improvements need to be made beforehand.

 Answer 1: Thank you very much for your valuable comments that led to the improvement of the manuscript.

Comment 2: Abstract: I would be very careful about relegating the origin of Wohlfathiimonas chitiniclastica infections to the tropics. The first reports derive from more temperate zones and it is unclear whether the bacterium is ubiquitous. The first description was made just over ten years ago ….

 Answer 2: We understand your concerns. The sentence was amended to read:

„Infections with parasitic larvae occur also in moderate climate (especially in people living in poor conditions) and, therefore, infection with rare bacteria associated with accidental myiasis, such as W. chitiniclastica, can be expected to become more common even there“

Comment 3:  Introduction: „In the last decade …“ please provide a reference here.

 Answer 3: The reference was added.

Comment 4: Is there evidence to which genus did the maggots belong? Was this examined? Please comment on this and evaluate why this may be an important information in myiasis and Wohlfathiimonas chitiniclastica related infections.

 Answer 4: This has been mentioned already in the first version of the manuscript as a limitation. Moreover, the part of the text highlighted below in bold was newly added.

„Despite the fact that W. magnifica is the most likely vector of this myiasis in the Czech Republic, another possible vector, a new invasive representative of the order Diptera, Clogmia albipunctata, has been also reported in the geographical area where our patient resided [16]. The identification of the vector has not been performed in our case. Although it is not important for clinical practice, it may be perceived as a limitation of this paper from the epidemiological perspective. If a particular vector overpopulation occurs in a certain area, we could potentially expect an increased risk of rare pathogen infections, including W. chitiniclastica. For this reason, we would also like to recommend the determination of the vector where possible in any future cases of myiasis, especially if a rare pathogen has been identified. This can be done, for example, by inserting the larvae into a fixation solution and if a rare pathogen is determined, cooperation with entomologists can be initiated.

Also, we added a sentence in the Recommendations:

„The larvae should be also preserved for potential future identification.“

Comment 5: Pathogen identification: Please mention the manufacturers of the culture media used here.

 Answer 5: The manufacturers were added to the respective media.

Comment 6: Pathogen identification: Why were the strains incubated at an elevated CO2 concentration?

 Answer 6: A standard set of culture media/conditions was used for the identification of pathogens in the wound (with blood agar being typically cultured in the CO2 atmosphere). We did not expect the occurrence of such a rare pathogen. Still, it is good to know that even a strictly aerobic bacteria such as W. chitiniclastica can be cultured in this way and, therefore, is covered by such a standard culture set.

The following was added in Discussion:

„It should be mentioned that W. chitiniclastica was cultured on blood agar in an elevated CO2 concentration, which falls within a standard set of culture conditions, and, therefore, no special culture is necessary“

Comment 7: Pathogen identification: „ …. were then identified using the MALDI TOF ….“ It should be „MALDI biotyper“ or more precise MALDI TOF MS. Please mention the exact unit designation and the versions of software and databases.

 Answer 7: Amended as requested.

Comment 8: Pathogen identification: „ … disc diffusion method and minimum inhibition concentrations (MIC)…“ Please mention in this (material and methods) section the manufactures and tests used.

 Answer 8: This is actually described in detail in the chapter 2.2. Antimicrobial susceptibility profile of W. chitiniclastica. However, we agree that in the first occurrence in the previous chapter, it is confusing and incomplete. For this reason, we have removed it from the previous chapter and kept the information only in the dedicated chapter 2.2.

Comment 9: Pathogen identification: Please indicate the sequence of the primers used.

 Answer 9: The information was added into the manuscript.

“...forward primer 5´-AYTGGGYDTAAAGNG-3´ and reverse primer 5´-CCGTCAATTYYTTTRAGTTT-3´ were used.”

Comment 10: Review and discussion: „In the past myiasis ….“ Please provide a reference here.

 Answer 10: Citation was added.

Comment 11: Review and discussion: Please speculate whether biochemical methods for identification of Wohlfathiimonas chitiniclastica are useful or not. There is data on VITEK 2 from biomerieux but hardly any data on other systems. Would a follow-up study investigatig other instruments be useful?

Answer 11: The following was added into the manuscript:

“As no data are available on the performance of other systems, a study evaluating the performance of other methods with respect to identification of W. chitiniclastica could be valuable for clinical practice.”

Comment 12: English language and style are fine/minor spell check required
Answer 12: Thank you.

Reviewer 2 Report

In the current review Martin Hladik and colleagues provided significant insight into Wohlfahrtiimonas chitiniclastica, a rare bacteria pathogen associated infection and methods for identification of this pathogen. Authors have explained isolation of W. chitiniclastica from a burn wound with accidental myiasis in a patient from Central Europe and a set of recommendations for care of a patient. The study design is straightforward, and it is a nicely written article.

Current review should be submitted as research article

The article’s title should be modified as, “first report of a burn-wound infection after accidental myiasis in Central Europe”

Microscopic view of the morphology of W. chitiniclastica is not clear. A clear microscopic image should be provided for better understanding.

A recent publication by Qi et al. on  W. chitiniclastica induced mouse infection should be mentioned.

Author Response

To: Reviewer 2

Comment 1: In the current review Martin Hladik and colleagues provided significant insight into Wohlfahrtiimonas chitiniclastica, a rare bacteria pathogen associated infection and methods for identification of this pathogen. Authors have explained isolation of W. chitiniclastica from a burn wound with accidental myiasis in a patient from Central Europe and a set of recommendations for care of a patient. The study design is straightforward, and it is a nicely written article.

 Answer 1: Thank you very much for your valuable comments that led to the improvement of the manuscript.

Comment 2: Current review should be submitted as research article. The article’s title should be modified as, “first report of a burn-wound infection after accidental myiasis in Central Europe”

 Answer 2: The title of our manuscript was changed according your suggestion.

Comment 3: Microscopic view of the morphology of W. chitiniclastica is not clearA clear microscopic image should be provided for better understanding.

 Answer 3: Unfortunately, our equipment at the time did not allow us to take better images of this. We propose to leave as is; however, if you insist, we can remove it from the Fig. 4.

Comment 4: A recent publication by Qi et al. on  W. chitiniclastica induced mouse infection should be mentioned.

Answer 4: Thank you, the citation was added.

Comment 5: Moderate English changes required.

Answer 5: The English of the manuscript was further improved by a professional agency.
